# UNIFIED AND EFFICIENT MULTI-VIEW CLUSTERING FROM PROBABILISTIC PERSPECTIVE

**Yalan Qin, Guorui Feng** *
School of Communication and Information Engineering, Shanghai University
{ylqin,grfeng}@shu.edu.edu

## ABSTRACT

Multi-view clustering aims to segment the view-specific data into the corresponding clusters. There have been a large number of works for multi-view clustering in recent years. As representative methods in multi-view clustering, works built on the graph make use of a view-consistent and discriminative graph while utilizing graph partitioning for the final clustering results. Despite the achieved significant success, these methods usually construct full graphs and the efficiency is not well guaranteed for the multi-view datasets with large scales. To handle the large-scale data, multi-view clustering methods based on anchor have been developed by learning the anchor graph with smaller size. However, the existing works neglect the interpretability of multi-view clustering based on anchor from the probabilistic perspective. These methods also ignore analyzing the relationship between the input data and the final clustering results based on the assigned meaningful probability associations in a unified manner. In this work, we propose a novel method termed Unified and Efficient Multi-view Clustering from Probabilistic perspective(UEMCP). It aims to improve the explanation ability of multi-view clustering based on anchor from the probabilistic perspective in an end-to-end manner. It ensures the consistent inherent structures among these views by learning the common transition probability from data points to categories in one step. With the guidance of the common transition probability matrix from data points to categories, the soft label of data points can be achieved based on the common transition probability matrix from anchor points to categories in the learning framework. Experiments on multi-view datasets confirm the superiority of UEMCP compared with the representative ones.

## 1 INTRODUCTION

As an important role in data mining and machine learning, clustering can be adopted to explore the organization and the latent structure of the data and then divide them into the corresponding clusters Qin et al. (2021; 2022; 2023d;a; 2024b; 2023c; 2025f); Pu et al. (2023); Qin et al. (2025d;g;b;h); Qin & Qian (2024); Qin et al. (2024c;d; 2025e; 2023b; 2025a;c; 2023e); Li et al. (2023b;c); Liu et al. (2024; 2023a; 2025); Lu et al. (2024); Zhang et al. (2025); Li et al. (2025); Liu et al. (2023b; 2022b;a). With the great progress in feature extraction and data generation techniques, the multi-view data has witnessed a significant rise. For instance, an image can be characterized with different feature representations, i.e., HOG, GIST and LBP. Multi-view clustering aims to group the data with multiple views into distinct clusters, which has been widely used in fields including object recognition and image segmentation. It is a paradigm in unsupervised learning, which is particular useful in the scenario when the explicit labels are not available.

There have been various developments for multi-view clustering domains Nie et al. (2017); Wang et al. (2019); Li et al. (2017); Luo et al. (2018); Wang et al. (2020b; 2024) in recent years, which include non-negative matrix factorization-based clustering methods Li et al. (2017), subspace-based clustering methods Luo et al. (2018) and graph-based clustering methods Wang et al. (2020b). As representative methods in multi-view clustering, works based on the graph utilize a view-consistent and discriminative graph while leveraging graph partitioning for clustering. These methods construct

---

*Corresponding author

the similarity graphs and calculate the eigen-decompose of Laplacian matrices. The computation costs of these two parts are $\mathcal{O}(n^2)$ and $\mathcal{O}(n^3)$, where $n$ denotes the size of dataset. This kind of methods has gained considerable attention due to their efficacy in discovering the complex hidden structures in the data and representing inter-data relationships.

Among the existing methods based on graph, Nie et al. Nie et al. (2017) adaptively tune the weights of graphs across views based on their contributions to clustering by introducing a non-parametric weight learning strategy, resulting in the enhanced performance. Wu et al. Wu et al. (2019) aim to achive superior view-consistent graph quality based on the combination of optimization function regarding the complementary information from different views and tensor nuclear norm constraints. Nie et al. Nie et al. (2018) facilitate the direct label extraction to achieve a view-consistent graph with the guidance of a Laplacian rank constraint built on the clear connected components. Despite the significant success obtained by these methods, they usually need to build $n \times n$ graphs and become inefficient for the multi-view datasets with large scales.

To address the above issues, multi-view clustering methods based on anchor have been presented for handling large-scale data by learning the bipartite graphs with $k$ connected components, where $k$ denotes the number of categories in the dataset. These methods usually select $m$ representative data points from the dataset as anchors, which is able to well balance the computational costs and global representation of the data. Li et al. Li et al. (2015) employ the pre-defined anchor graphs with high quality to realize multi-view clustering on large-scale datasets. Xia et al. Xia et al. (2023) effectively harness the complementary information from different views and guarantee the view consistency by using Schatten $p$-norm regularization on the tensor constructed by bipartite graphs. Li et al. Li et al. (2022) aim to achieve clear component segregation in the bipartite graphs by imposing Laplacian rank constraints in the bipartite graph learning. You et al. You et al. (2023) introduce predefined anchor point labels and constraints on anchor points to obtain discriminative cluster structures, resulting in superior bipartite graphs for clustering. However, these methods ignore the interpretability of anchor-based multi-view clustering from the probabilistic perspective, which inevitably complicate the acquirement of $k$ distinct connected components. They also fail to analyze the relationship between the input data and the final clustering results by assigning meaningful probability associations in a unified manner.

To deal with the above issues, we propose a novel method termed Unified and Efficient Multi-view Clustering from Probabilistic perspective(UEMCP). This method aims to increase the explanation ability of multi-view clustering based on anchor from the probabilistic perspective in an end-to-end manner. It is able to learn the common transition probability from data points to categories shared by different views with one step, which guarantees the consistent inherent structures among these views. By introducing the common transition probability matrix from anchor points to categories, we can achieve the soft labels of data points with the guidance of the common transition probability matrix from data points to categories in the learning procedure. To be specific, the discrepancy between the obtained soft labels and the common transition probability from data points to categories based on the Frobenius norm is calculated as the loss function to enhance the reliability. The main contributions in our work are summarized as:

1. We propose a novel method termed Unified and Efficient Multi-view Clustering from Probabilistic perspective (UEMCP), which assigns the probabilistic meaning to the anchor graph and soft label of data points to increase the explanation ability of multi-view clustering model in an end-to-end manner.

2. UEMCP learns the common transition probability from data points to categories shared by multiple views with one step, which is able to ensure the consistency of inherent structures for these views. Based on the common transition probability matrix from anchor points to categories, the soft labels of data points can be achieved with the guidance of the common transition probability matrix from data points to categories in the learning framework.

3. Extensive experiments are performed on several datasets to validate the effectiveness and efficiency of the proposed UEMCP under different metrics.

## 2 RELATED WORKS

### 2.1 MULTI-VIEW CLUSTERING

In multi-view clustering process, we can obtain the final clustering outcomes by constructing the similarity graph and then run the existing clustering algorithm on this graph. Gao et al. Gao et al. (2020) combine dimension reduction, feature selection and manifold structure learning to realize multi-view clustering. It is able to reduce the effects of the existing redundancy and noise by mapping the data with high dimension to the low-dimensional spaces. Cao et al. Cao et al. (2015) use the Hilbert Schmidt Independence Criterion (HSIC) to guarantee the diversity of multiple subspace representations. The complementary among different views is further explored in this manner. Wang et al. Wang et al. (2020a) simultaneously suppress noise, reduce dimension and learn the local structure graph based on $L_{2,1}$-norm in achieving the final clustering results. The optimal graph is able to be directly obtained and the extra processing steps are not needed in the whole procedure. Gao et al. Gao et al. (2015) adopt a shared indicator matrix to integrate representation matrices across views, which consider the magnitudes of these representations. Sang et al. Sang et al. (2022) learn consensus structure graph, consider manifold structure and reduce dimension by auto-weighted multi-view projection based on graph. The similarity graph is built on the projected data to ensure that the noisy and redundant information is removed. Wang et al. Wang et al. (2017) adopt an item related to the position for measuring the diversities among different representations and ensure that an indicator is shared by these representations by exploring the consistency. Wang et al. Wang et al. (2022a) analyze the multi-view dataset and learn the shared projection matrix based on the diversity and consistency preserving. Wang et al. Wang et al. (2015) can better handle the noise corruption by designing an angular regularizer and refining it based on the sparse decomposition. Li et al. Li et al. (2023a) build an orthogonal projection matrix to achieve the feature representation for each view and learn the representation in an embedded space. Luo et al. Luo et al. (2018) jointly explore the consistency and specificity of different views based on a shared representation and particular representations. The above methods are able to achieve promising clustering results. However, these methods usually need relatively high computational complexity and space complexity, which inevitably hamper their further applications.

### 2.2 ANCHOR GRAPH-BASED MULTI-VIEW CLUSTERING

To decrease the complexity of multi-view clustering, the approaches based on anchor graph have been extensively presented Kang et al. (2020). The anchor graph can be obtained by constructing the relationship between $m$ anchors and $n$ data points, which is able to cover $n$ data points with $m$ anchors. Due to $m \ll n$, the anchor graph-based multi-view clustering methods can effectively deal with datasets with large scales. For instance, Zhan et al. Zhan et al. (2018) utilize $K$ connected components rather than rely on post-processing in extracting clustering metrics and a global graph can be constructed from various single view graphs. Guo et al. Guo & Ye (2019) build up the similarity relation between data points and anchors and then integrate the relations within the view as well as among views into the fused similarity matrix. Li et al. Li et al. (2022) fuse cluster graphs from different views based on a parameter-free method, which leads to composite graphs by utilizing self-supervised weighting. Kang et al. Kang et al. (2020) construct smaller graphs for all views and then employ the integration mechanism to accelerate the eigen-decomposition. Zhou et al. Zhou et al. (2022) minimize nuclear norm and tensor Schatten $p$-norm in considering the inter-view and intra-view spatial low-rank structures of the bipartite graphs. Huang et al. Huang et al. (2020) assign weights to individual graphs by learning and the auxiliary parameters are not needed. Yang et al. Yang et al. (2022) explore the synergistic information among different views based on the Schatten $p$-norm. Yu et al. Yu et al. (2022) converse the similarity learning to the consensus similarity construction.

Even though the above multi-view clustering methods have achieved preferable performance, these still neglect the interpretability from the probabilistic perspective for the anchor-based multi-view clustering, complicating the acquirement of $k$ distinct connected components. These methods also pay few attentions to analyzing the relationship between the input data and the final clustering results in a unified manner.

## 3 METHODOLOGY

In this part, we present the inspiration and formulation of UEMCP, the optimization details and the analysis of computation complexity for the proposed method.

### 3.1 INSPIRATION AND FORMULATION

Most existing multi-view clustering methods for large-scale data usually rely on anchor graphs to reduce the algorithmic complexity. The structure of the data can be represented by choosing $m$ anchors to reflect the entire distribution for the dataset. The relation between data points and anchors can be built based on the anchor graph and the correlation tends to be stronger when a data point and anchor belong to the same category. The anchor graph can be considered as the probability transition matrix between data points and anchors due to the non-negative properties and summarization being one for the row. The anchor selection and the probability transition matrix construction are separated from each other, which will inevitably affect the final performance. Different from the traditional strategy, we automatically learn anchors instead of simple sampling. Besides, a consensus data distribution shared by different views is assumed in our multi-view clustering setting. We then define the projection matrix $\{P_v\}_{v=1}^V$ to realize the consensus anchor goal with dimension being $d_v \times d$ as follows:

$$\min_{P_v, A, S} \sum_{v=1}^V \alpha_v^2 \|X_v - P_v AS\|_F^2,$$
$$s.t. \ P_v^T P_v = I_d, \ A^T A = I_m, \ S \geq 0, \ S^T 1 = 1, \ \alpha^T 1 = 1,$$

(1)

where $m$ represents the number of anchors, $X_v \in R^{d_v \times n}$ is the data of the $v$-th view, $n$ indicates the number of data points, $A$ denotes the unified anchor matrix and $S \in R^{m \times n}$ refers to the corresponding anchor graph. $S \geq 0$ and $S^T 1 = 1$ are adopted in the optimization function, which can be viewed as the probability transition matrix between data points and anchors.

Based on the consensus anchor $A \in R^{d \times m}$, the transition probability matrix from anchors to categories $H \in R^{c \times m}$ and the transition probability matrix from data points to categories $S \in R^{m \times n}$ can be calculated. The optimization function is shown as follows:

$$\min_G \|HS - G\|_F^2, \ s.t. \ G \geq 0, \ G^T 1 = 1,$$

(2)

where $G \in R^{c \times n}$ denotes the probability transition matrix between data points and categories. $H \in R^{c \times m}$ represents the probability transition matrix between anchors and categories. We then combine the above two parts and rewrite the objective function as:

$$\min_{P_v, A, S, H, G} \sum_{v=1}^V \alpha_v^2 \|X_v - P_v AS\|_F^2 + \lambda \|HS - G\|_F^2,$$
$$s.t. \ P_v^T P_v = I_d, \ A^T A = I_m, \ S \geq 0, \ S^T 1 = 1, \ H \geq 0,$$
$$H^T 1 = 1, G^T 1 = 1, \ G \geq 0, \ \alpha^T 1 = 1,$$

(3)

where $\lambda > 0$ is the parameter to balance different terms. Since $G$ is adopted to achieve the labels of data points, we relax the constraints imposed on $G$ to be orthogonal and non-negative for more intuitive understanding of the category. Then each row in $G$ just contains a non-zero value and the corresponding position represents the label of the data point. Therefore, the final objective function is formulated as:

$$\min_{P_v, A, S, H, G} \sum_{v=1}^V \alpha_v^2 \|X_v - P_v AS\|_F^2 + \lambda \|HS - G\|_F^2,$$
$$s.t. \ P_v^T P_v = I_d, \ A^T A = I_m, \ S \geq 0, \ S^T 1 = 1, \ H \geq 0,$$
$$H^T 1 = 1, \ G^T G = I_m, \ G \geq 0, \ \alpha^T 1 = 1.$$

(4)

The advantages of the above proposed method are summarized as follows:

1. **Adaptive weighted**: Different from some existing multi-view clustering methods based on anchor, UEMCP adaptively learns the coefficient of each view based on their contributions to the consensus anchor graph.

2. **Probabilistic**: UEMCP assigns the probabilistic meaning to the anchor graph and soft label of data points to increase the explanation ability of multi-view clustering model.

3. **Consistent**: Despite the differences in the distributions for different views, the intrinsic structure of the data among these views is able to be consistent with the guidance of the introduced common transition probability matrix from anchor points to categories.

## 3.2 OPTIMIZATION

To deal with the optimization problem in Eq. (4), we give an alternate optimization algorithm in finding the solution to each variable with the others being fixed.

$S$-**subproblem**: When the other variables are fixed, the objective function regarding $S$ is written as

$$\min_S \sum_{v=1}^{V} \alpha_v^2 \|X_v - P_v A S\|_F^2 + \lambda \|HS - G\|_F^2, \ s.t. \ S \geq 0, \ S^T 1 = 1. \tag{5}$$

We transform the above optimization problem as the Quadratic Programming (QP) in the following:

$$\min_S \frac{1}{2} S_{:,j}^T W S_{:,j} + f^T S_{:,j}, \ s.t. \ S_{:,j}^T 1 = 1, \ S \geq 0, \tag{6}$$

where $W = 2(\sum_{v=1}^{V} \alpha_v^2 + \lambda)I$, $f^T = -2\sum_{v=1}^{V} X_{[:,j]}^T P_v A - 2\lambda G_{[:,j]}^T H$. We then solve the QP problem for each row of $S$ in the optimization process.

$A$-**subproblem**: When the other variables are fixed, the objective function regarding $A$ is written as

$$\min_A \sum_{v=1}^{V} \alpha_v^2 \|X_v - P_v A S\|_F^2, \ s.t. \ A^T A = I. \tag{7}$$

We rewrite the optimization problem of $A$ in the above as follows:

$$\max_A Tr(A^T C), \ s.t. \ A^T A = I, \tag{8}$$

where $C = \sum_{v=1}^{V} \alpha_v^2 P_v^T X_v S^T$. Assuming the singular value decomposition (SVD) of $A$ is $U_A \Sigma_A V_A^T$, we achieve the optimal $A$ by calculating $U_A V_A^T$.

$P_v$-**subproblem**: When the other variables are fixed, the objective function regarding $P_v$ is written as

$$\min_{P_v} \sum_{v=1}^{V} \alpha_v^2 \|X_v - P_v A S\|_F^2, \ s.t. \ P_v^T P_v = I. \tag{9}$$

We then transform the above problem as follows:

$$\max_{P_v} Tr(P_v^T B_v), \ s.t. \ P_v^T P_v = I, \tag{10}$$

where $B_v = X_v S^T A^T$. Supposing the SVD of $P_v$ is $U_{P_v} \Sigma_{P_v} V_{P_v}^T$, the optimal $P_v$ can be achieved by calculating $U_{P_v} V_{P_v}^T$.

$H$-**subproblem**: When the other variables are fixed, the objective function regarding $H$ is written as

$$\min_H \|HS - G\|_F^2, \ s.t. \ H \geq 0, \ H^T 1 = 1. \tag{11}$$

The above objective function can be formulated by the minimization problem as follows:

$$\min_{H_{:,j}} \|H_{:,j} S - G_{:,j}\|^2, \ s.t. \ H \geq 0, \ H^T 1 = 1. \tag{12}$$

The optimal row of the above formulation is

$$i^* = \arg \min_i \|G_{:,j} - S_{:,i}\|^2. \tag{13}$$

Then the optimal cluster assignment is achieved by minimizing the above optimization problem.

$G$-**subproblem**: When the other variables are fixed, the objective function regarding $G$ is written as

$$\min_G \|HS - G\|_F^2, \ s.t. \ G^T G = I_m, \ G \geq 0. \tag{14}$$

Likewise, the minimization of $G$ is formulated as:

$$\max_G Tr(G^T J), \ s.t. \ G^T G = I_m, \ G \geq 0, \tag{15}$$

where $J = HS$. Therefore, the optimal $G$ is $U_J V_J^T$ with $J$ being $U_J \Sigma_J V_J^T$.

$\boldsymbol{\alpha_v}$-**subproblem**: With the other variables being fixed, the objective function regarding $\alpha_v$ is written as

$$\min_\alpha \sum_{v=1}^V \alpha_v^2 \|X_v - P_v AS\|_F^2, \ s.t. \ \alpha^T \mathbf{1} = 1. \tag{16}$$

Then the optimal $\alpha_v$ is achieved according to Cauchy-Buniakowsky-Schwarz inequality as follows:

$$\alpha_v = \frac{\frac{1}{\|X^v - P^v AS\|_F}}{\sum_{v=1}^V \frac{1}{\|X^v - P^v AS\|_F}}. \tag{17}$$

Due to the convex property in each sub-problem, the objective function monotonically decreases in each iteration until convergence. The detailed process in solving the proposed UEMCP is shown in Algorithm 1.

---

**Algorithm 1:** Algorithm of UEMCP

**Input:** Dataset $\{X_v\}_{v=1}^v$, number of clusters $k$.
**Output:** Probability transistion matrix between data points and categories $G$.
**Initialize:** Initialize $A$, $P_v$, $\{\alpha_v\}_{p=1}^v$, $S$, $H$ and $G$.
**repeat**
    Update $S$ by solving Eq. (6);
    Update $\{P_v\}_{p=1}^v$ by solving Eq. (10);
    Update $A$ by solving Eq. (8);
    Update $G$ by solving Eq. (15);
    Update $H$ by solving Eq. (13);
    Update $\alpha$ by solving Eq. (17);
**until** *convergence*;

---

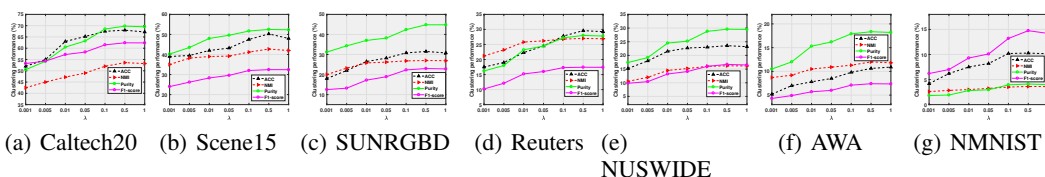

(a) Caltech20  (b) Scene15  (c) SUNRGBD  (d) Reuters  (e) NUSWIDE  (f) AWA  (g) NMNIST

Figure 1: Parameter study of $\lambda$ on all datasets.

## 3.3 COMPLEXITY ANALYSIS

The total computational complexity of the proposed UEMCP contains the optimization cost of all variables. It needs $\mathcal{O}(d_v d^2)$ to conduct SVD and $\mathcal{O}(d_v dk^2)$ to achieve the optimal $P_v$ by performing matrix multiplication, where $d = \sum_{v=1}^V d_v$. Likewise, the time cost of updating $A$ includes $\mathcal{O}(dm^2)$ for SVD and $\mathcal{O}(dmk^2)$ in matrix multiplication. Besides, solving $S$ consumes $\mathcal{O}(nm^3)$ for all data vectors. It costs $\mathcal{O}(mk^2)$ and $\mathcal{O}(mk^3)$ for $H$ in SVD and matrix multiplication, respectively. For updating $G$, the corresponding time cost is $\mathcal{O}(mnk)$. It just needs $\mathcal{O}(1)$ in calculating $\alpha_v$. Therefore, the computational complexity of UEMCP is $\mathcal{O}(n)$ regarding the number of data points.

Table 1: Clustering results based on ACC (%) on all datasets. "N/A " denotes out of memory.

| Dataset | ETLMSC | SFMC | BMVC | LMVSC | FMCNOF | FPMVS | MSC-BG | EDMC | Ours |
|---------|--------|------|------|-------|--------|-------|--------|------|------|
| Caltech101-20 | 48.95±0.20 | 59.50±0.05 | 16.95±0.01 | 29.20±0.50 | 49.50±0.00 | 66.20±0.00 | 64.80±0.20 | 65.10±0.05 | **68.00±0.20** |
| Scene15 | 20.50±0.05 | 22.08±0.01 | 29.80±0.00 | 30.75±0.00 | 24.20±0.00 | 40.50±0.00 | 45.40±0.00 | 48.00±0.50 | **50.27±0.15** |
| SUNRGBD | 10.20±0.00 | 11.58±0.02 | 17.00±0.50 | 18.65±0.00 | 13.20±0.05 | 23.75±0.01 | 26.00±0.00 | 27.00±0.00 | **31.50±0.00** |
| Reuters | 11.50±0.10 | 12.00±0.00 | 15.90±0.20 | 16.20±0.00 | 12.00±0.25 | 20.58±0.00 | 23.65±0.10 | 25.27±0.00 | **29.60±0.00** |
| NUSWIDEOBJ | N/A | 12.95±0.05 | 13.72±0.01 | 15.20±0.00 | 12.00±0.01 | 19.54±0.00 | 19.75±0.15 | 21.10±0.00 | **23.56±0.05** |
| AWA | N/A | 3.95±0.05 | 8.72±0.01 | 7.40±0.05 | 8.25±0.01 | 9.00±0.00 | 8.80±0.00 | 9.15±0.05 | **10.62±0.00** |
| NoisyMNIST | N/A | N/A | 5.29±0.00 | 5.15±0.10 | 6.20±0.00 | 7.58±0.00 | 7.60±0.00 | 9.00±0.10 | **10.28±0.05** |

Table 2: Clustering results based on NMI (%) on all datasets. "N/A " denotes out of memory.

| Dataset | ETLMSC | SFMC | BMVC | LMVSC | FMCNOF | FPMVS | MSC-BG | EDMC | Ours |
|---------|--------|------|------|-------|--------|-------|--------|------|------|
| Caltech101-20 | 45.26±0.05 | 43.15±0.00 | 16.46±0.05 | 41.80±0.00 | 31.47±0.00 | **63.28±0.01** | 51.05±0.00 | 52.00±0.05 | 53.60±0.00 |
| Scene15 | 32.50±0.00 | 30.18±0.10 | 13.00±0.00 | 30.25±0.10 | 22.64±0.00 | **45.70±0.15** | 39.46±0.00 | 40.00±0.05 | 42.70±0.10 |
| SUNRGBD | 18.75±0.05 | 2.26±0.00 | 19.37±0.01 | 25.43±0.10 | 9.45±0.01 | 22.28±0.05 | 23.00±0.01 | 24.68±0.10 | **27.00±0.05** |
| Reuters | 12.90±0.00 | 2.58±0.00 | 18.20±0.00 | 20.50±0.00 | 9.80±0.00 | 25.20±0.00 | 24.37±0.05 | 28.00±0.05 | **29.95±0.00** |
| NUSWIDEOBJ | N/A | 0.95±0.05 | 12.85±0.01 | 12.75±0.15 | 5.82±0.01 | 13.27±0.01 | 13.15±0.00 | 14.70±0.02 | **16.20±0.00** |
| AWA | N/A | 0.32±0.01 | 13.56±0.00 | 8.62±0.01 | 7.95±0.00 | 10.46±0.10 | 9.82±0.01 | 10.15±0.00 | **12.00±0.10** |
| NoisyMNIST | N/A | N/A | 6.85±0.01 | 12.68±0.05 | 1.08±0.01 | 2.59±0.05 | 2.80±0.00 | 3.05±0.15 | **3.62±0.00** |

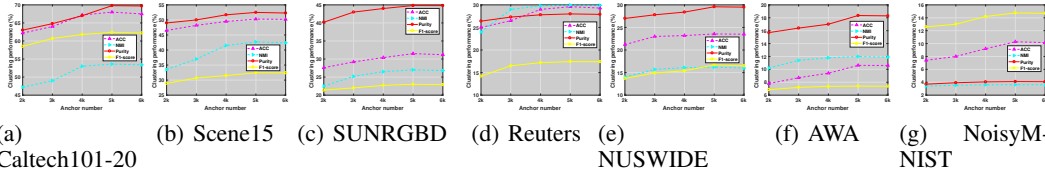

| (a) Caltech101-20 | (b) Scene15 | (c) SUNRGBD | (d) Reuters | (e) NUSWIDE | (f) AWA | (g) NoisyM-NIST |

Figure 2: Sensity investigation on all datasets.

# 4 EXPERIMENTS

We validate the proposed UEMCP on seven widely adopted multi-view datasets by performing extensive experiments based on the clustering performance and computational efficiency in this section. The memory of the adopted device for running experiment is 8G.

## 4.1 DATASETS

The proposed UEMCP is evaluated on seven multi-view datasets and we show the details of these datasets on Table 1. Caltech101-20 consists of 2386 data points from 20 categories, which is a subset of the collection Caltech101. Scene15 contains total three views and 4485 instances in 15 categories. SUNRGBD has 45 classes and 10335 indoor scene images from two views. Reuters consists of 18758 data points and five views over six classes. NUSWIDEOBJ is the object image dataset, which includes 30000 images from 31 classes. AwA is an animal dataset with attributes, composed of 50 kinds of animals in six features. NoisyNMIST has 50000 samples and two views in 10 categories.

## 4.2 COMPARED METHODS AND EXPERIMENTAL SETTINGS

The proposed UEMCP is compared with the eight multi-view clustering works shown in the following. **BMVC Zhang et al. (2019)** reduces the computational complexity by concurrently learning the consensus clustering result and multi-view binary representation. **LMVSC Kang et al. (2020)** constructs a consistent bipartite graph based on a multi-graph fusion strategy and determines the cluster labels by spectral clustering. **SFMC Li et al. (2022)** uses a bipartite graph learning strategy with Laplace rank constraints to build view-consistent bipartite graphs. **ETLMSC Wu et al. (2019)** applies spectral clustering to the constructed probability matrix from a tensor, resulting in the final results. **FMCNOF Yang et al. (2021)** adopts the anchor selection and non-negative matrix factorization based on a fast clustering method. **FPMVS Wang et al. (2022b)** achieves joint optimization by combining graph construction and anchor learning in a framework. **MSC-BG Yang et al. (2022)** constrains the bipartite graph with the Scatten $p$-norm to effectively capture the complementary in-

Table 3: Clustering results based on Purity (%) on all datasets. "N/A " denotes out of memory.

| Dataset | ETLMSC | SFMC | BMVC | LMVSC | FMCNOF | FPMVS | MSC-BG | EDMC | Ours |
|---|---|---|---|---|---|---|---|---|---|
| Caltech101-20 | 71.20±0.05 | 65.00±0.20 | 42.50±0.05 | 65.78±0.00 | 52.35±0.10 | 66.20±0.00 | 59.57±0.18 | 61.58±0.00 | **69.79±0.10** |
| Scene15 | 50.60±0.00 | 42.85±0.50 | 45.80±0.00 | 49.20±0.00 | 41.35±0.50 | 48.20±0.00 | 46.40±0.00 | 47.89±0.00 | **52.50±0.00** |
| SUNRGBD | 15.20±0.00 | 10.58±0.05 | 33.60±0.05 | 38.20±0.00 | 34.00±0.05 | 34.47±0.50 | 39.52±0.00 | 42.30±0.00 | **44.82±0.10** |
| Reuters | 11.46±0.00 | 9.82±0.15 | 15.20±0.00 | 18.00±0.00 | 10.35±0.05 | 23.19±0.00 | 22.50±0.00 | 26.46±0.00 | **28.00±0.00** |
| NUSWIDEOBJ | N/A | 13.20±0.50 | 21.75±0.00 | 22.60±0.10 | 24.00±0.00 | 23.57±0.05 | 26.28±0.00 | 27.00±0.00 | **29.58±0.00** |
| AWA | N/A | 3.40±0.17 | 8.42±0.00 | 10.20±0.00 | 11.85±0.15 | 13.59±0.00 | 15.47±0.00 | 17.00±0.01 | **18.35±0.00** |
| NoisyMNIST | N/A | N/A | 2.70±0.00 | 2.85±0.10 | 3.09±0.07 | 3.27±0.00 | 3.76±0.20 | 3.95±0.00 | **4.10±0.00** |

Table 4: Clustering results based on F1-score (%) on all datasets. "N/A " denotes out of memory.

| Dataset | ETLMSC | SFMC | BMVC | LMVSC | FMCNOF | FPMVS | MSC-BG | EDMC | Ours |
|---|---|---|---|---|---|---|---|---|---|
| Caltech101-20 | 44.30±0.05 | 52.85±0.00 | 48.50±0.00 | 37.69±0.00 | 49.20±0.00 | 48.58±0.15 | 48.69±0.10 | 58.46±0.00 | **62.40±0.00** |
| Scene15 | 25.20±0.00 | 23.50±0.10 | 27.00±0.00 | 28.48±0.00 | 29.00±0.05 | 27.18±0.00 | 28.65±0.05 | 30.00±0.02 | **32.45±0.00** |
| SUNRGBD | 16.90±0.00 | 15.20±0.05 | 18.53±0.00 | 12.46±0.10 | 14.00±0.05 | 15.84±0.00 | 17.98±0.00 | 20.40±0.00 | **23.00±0.05** |
| Reuters | 12.00±0.00 | 11.46±0.10 | 12.52±0.00 | 14.20±0.05 | 14.59±0.00 | 12.70±0.00 | 15.00±0.10 | 15.28±0.00 | **17.43±0.00** |
| NUSWIDEOBJ | N/A | 11.52±0.00 | 10.48±0.00 | 9.96±0.15 | 8.80±0.00 | 9.80±0.00 | 11.50±0.10 | 14.27±0.00 | **16.70±0.00** |
| AWA | N/A | 4.37±0.00 | 3.50±0.00 | 5.52±0.00 | 4.90±0.00 | 3.78±0.00 | 5.90±0.00 | 6.27±0.01 | **7.40±0.00** |
| NoisyMNIST | N/A | N/A | 5.95±0.01 | 6.28±0.00 | 8.40±0.19 | 12.58±0.00 | 13.90±0.00 | 14.20±0.00 | **14.75±0.00** |

formation and spatial structure among the views. **EDMC Qin et al. (2024a)** learns anchors from instance and cluster level in the row and column space of the feature representations across views.

To eliminate the randomness in initialization, we run $K$-means for 50 times in achieving the clustering results. For fair comparison, the parameters in the compared methods are set to be the same as the ones in their original literatures. We then evaluate the clustering performance by four widely adopted metrics, i.e., ACC, NMI, Purity and F1-score. To ensure the precision for the experimental results, we report the mean and variance from 20 iterations as the final results.

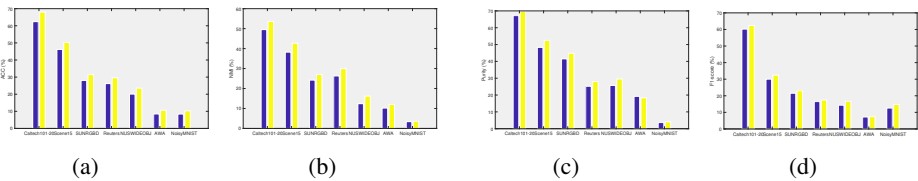

Figure 3: Ablation study on all datasets.

## 4.3 Influence of parameter $\lambda$

To study the influence of parameter $\lambda$ on the final clustering performance, we perform experiments for investigation in the range of $\{0.001, 0.005, 0.01, 0.05, 0.1, 0.5, 1\}$. According to Fig. 1, we find that choosing the proper parameter is crucial in improving the final clustering results. The parameter $\lambda$ with too large or small values is not helpful in achieving the desired performance. We can find that the stable performance can be obtained when $\lambda$ is in the range of $\{0.1, 0.5\}$. Besides, the best performance is achieved when $\lambda = 0.5$ for the proposed UEMCP.

## 4.4 Experimental results and Analysis

We present the clustering results of the proposed UEMCP and compared methods on seven benchmark datasets. 'N/A' is adopted to show that the method encounters the out-of-memory issue on the dataset. According to Tables 2-5, we can draw conclusions as follows:

- Compared with the existing multi-view clustering methods based on subspace, the anchor works are more proper for the dataset with large scales and can achieve better performance on most cases, demonstrating the necessity of adopting anchor graph.
- Some methods employ a two-stage clustering manner, i.e., ETLMSC, which will lead to the memory issues for datasets with large scales. In contrast, the proposed UEMCP performs anchor selection to mitigate such issues, especially for the large-scale clustering tasks.

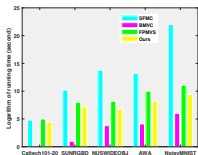

Figure 4: Logarithm of running time on different datasets.

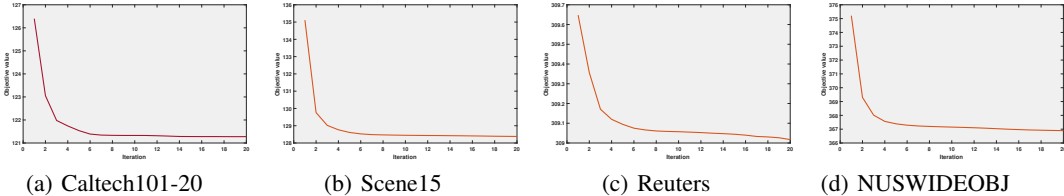

(a) Caltech101-20        (b) Scene15        (c) Reuters        (d) NUSWIDEOBJ

Figure 5: Convergence curves on different datasets.

- UEMCP shows obvious advantages over other compared methods under different metrics. For example, UEMCP improves 2.70% over EDMC on Scene15 in terms of NMI, which can be explained by that assigning the probabilistic meaning to the anchor graph and soft label of data points for increasing the explanation ability of multi-view clustering in end-to-end manner is necessary.

### 4.5 SENSITY INVESTIGATION

In this part, we study the influence of the anchor number for the final clustering results. We vary the anchor number in the range of $\{2k, 3k, 4k, 5k, 6k\}$ and investigate the sensity in terms of different metrics, where $k$ is the number of clusters in the dataset. According to Fig. 2, we observe that more anchor number can help lead to better performance and too large anchor number is not necessary for the final results, i.e., the better clustering performance can be achieved when anchor number is equal to $5k$ on NUSWIDEOBJ.

### 4.6 ABLATION STUDY

To study the significance of the regularization term in the final objective function, we perform ablation study by removing the second term in validation. According to Fig. 3, we find that the performance significantly drops without considering the probability transition matrix between data points and categories. It is because the soft label of data points can be achieved with the guidance of the common transition probability matrix from data points to categories in the learning procedure.

### 4.7 RUNNING TIME ANALYSIS AND CONVERGENCE ANALYSIS

To validate the computational efficiency, we list the running time of our UEMCP on different datasets. As shown in Fig. 4, we observe that UEMCP needs relatively shorter running time on different datasets, which demonstrates the computational efficiency of our method. As previously demonstrated, the time complexity of UEMCP is nearly to $\mathcal{O}(n)$. Though less running time is needed by BMVC, the simple procedure ignores to fully explore the information from multiple views, leading to relatively poor performance. We perform experiments to demonstrate the convergence of the proposed UEMCP on benchmark multi-view datasets. According to Fig. 5, we find that the objective values of UEMCP gradually decrease for each iteration and converge in about 20 iterations, which clearly show the convergence of the proposed UEMCP.

## 5 CONCLUSION

We propose a novel method termed Unified and Efficient Multi-view Clustering from Probabilistic perspective(UEMCP) in this paper, which can improve the explanation ability of multi-view clustering based on anchor from the probabilistic perspective in an end-to-end manner. UEMCP learns the common transition probability from data points to categories in one step, ensuring the consistency of inherent structures for these views. The alternate minimizing algorithm is given to solve the formulated problem. Extensive experiments illustrate the effectiveness and efficiency of UEMCP on seven multi-view datasets under four metrics.

### ACKNOWLEDGMENTS

This work was supported by Eastern Talent Plan Leading Project under Grant BJKJ2024011 and National Natural Science Foundation of China (62402303).

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
