# OpenReview forum: "Unified and Efficient Multi-view Clustering from Probabilistic Perspective"
_ICLR.cc/2026/Conference — ICLR 2026 Poster_

### Official Review · Reviewer_1Zmv · 2025-10-22

**Soundness:** 3
**Presentation:** 3
**Contribution:** 3
**Rating:** 8
**Confidence:** 5

**Summary:**

This paper proposes a novel method termed Unified and Efficient Multi-view Clustering from Probabilistic perspective (UEMCP). It aims to improve the explanation ability of multi-view clustering based on anchor from the probabilistic perspective in an end-to-end manner. It ensures the consistent inherent structures among these views by learning the common transition probability from data points to categories in one step. With the guidance of the common transition probability matrix from data points to categories, the soft label of data points can be achieved based on the common transition probability matrix from anchor points to categories in the learning framework.

**Strengths:**

The proposed method, which proposes a novel method termed Unified and Efficient Multi-view Clustering from Probabilistic perspective. It aims to improve the explanation ability of multi-view clustering based on anchor from the probabilistic perspective in an end-to-end manner. The soft label of data points can be achieved based on the common transition probability matrix from anchor points to categories in the learning framework.

**Weaknesses:**

1. The authors summarize the advantages of the proposed method based on the Adaptive weighted, Probabilistic and Consistent property. However, there is no further explanation for Probabilistic property in illustrating the advantages of the proposed UEMCP. The authors should add the related further explanation for Probabilistic property in the paper.
2. The second best results in Tables for the experiment part can be highlighted to make the authors more obviously grasp the clustering performance gains.
3. The authors do not give detailed analysis regarding the convergence study of the proposed UEMCP. More analysis can be given for the convergence study part in the experiment, i.e., the iteration speed for the proposed UEMCP in the paper.
4. The authors study the influence of the anchor number for the final clustering results in the experiment. They vary the anchor number in the range of {2k, 3k, 4k, 5k, 6k} and investigate the sensitivity in terms of different metrics, where k is the number of clusters in the dataset. The authors are expected to give the reason why they choose {2k, 3k, 4k, 5k, 6k} as the range in the experiment.
5. The authors are expected to adjust the size of the presentation for Algorithm 1 of the proposed method in the paper.

**Questions:**

The authors are expected to give the reason why they choose {2k, 3k, 4k, 5k, 6k} as the range in the experiment for the proposed method.

---

> ### Author Response · Authors · 2025-11-17
>
> Q1: The authors can add related further explanation for probabilistic property in this paper.
>
> A1: Thanks for the comment! As reviewer mentioned, we summarize the advantages of the proposed method based on the Adaptive weighted, Probabilistic and Consistent property. However, there is no further explanation for Probabilistic property in illustrating the advantages of the proposed UEMCP. Therefore, we should add the related further explanation for Probabilistic property in the paper. UEMCP assigns the probabilistic meaning to the anchor graph and soft label of data points by imposing non-negative and sum-to-one constraints to these two variables for increasing the explanation ability of multi-view clustering model. Then the anchor graph and soft label of data points in the proposed UEMCP own the probabilistic property in this manner. We will add such explanations regarding the probabilistic property for the camera-ready version.
>
> Q2: The second best results in Tables for the experiment part are expected to be highlighted.
>
> A2: Good question! We should highlight the second best results in Tables for the experiment part. We will highlight the second best results in Tables for the experiment part to make the authors more obviously grasp the clustering performance gains for the camera-ready version.
>
> Q3: The authors can give more analysis for the convergence study part.
>
> A3: Thanks for the comment! As reviewer mentioned, we do not give detailed analysis regarding the convergence study of the proposed UEMCP. More analysis can be given for the convergence study part in the experiment, i.e., the convergence speed for the proposed UEMCP in the paper. The convergence speed of the achieved curves for the proposed UEMCP on datasets is very fast and reach to relatively stable within some iterations. We will add the above analysis for the camera-ready version.
>
> Q4: The reason why the authors choose {2k, 3k, 4k, 5k, 6k} as the range in the experiment.
>
> A4: Good question! We study the influence of the anchor number for the final clustering results in the experiment. We vary the anchor number in the range of {2k, 3k, 4k, 5k, 6k} and investigate the sensitivity in terms of different metrics, where k is the number of clusters in the dataset. Therefore, we are expected to give the reason why we choose {2k, 3k, 4k, 5k, 6k} as the range in the experiment. The reason why we choose the range of {2k, 3k, 4k, 5k, 6k} in parameter selection is that it can guide the anchors to be uniformly produced in sample clusters, which avoids the situation that there are no anchors in sample clusters or the anchors are produced outside the clusters. The above explanation of anchor number selection is also demonstrated by the existing works, i.e., [a]. We will add such illustration regarding anchor number selection for the camera-ready version.
>
> [a] Anchor Learning with Potential Cluster Constraints for Multi-view Clustering, AAAI, 2025.
>
> Q5: The size of presentation for Algorithm 1 can be adjusted.
>
> A5: Thanks for the comment! As reviewer mentioned, we are expected to adjust the size of the presentation for Algorithm 1 of the proposed method in the paper. We will adjust the size of the presentation for Algorithm 1 to make it more compact in the camera-ready version.

---

### Official Review · Reviewer_ixUY · 2025-10-31

**Soundness:** 3
**Presentation:** 3
**Contribution:** 2
**Rating:** 6
**Confidence:** 4

**Summary:**

This paper proposes a method termed Unified and Efficient Multi-view Clustering from Probabilistic perspective (UEMCP), which assigns the probabilistic meaning to the anchor graph and soft label of data points to increase the explanation ability of multi-view clustering model in an end-to-end manner. UEMCP learns the common transition probability from data points to categories shared by multiple views with one step, which is able to ensure the consistency of inherent structures for these views. Experiments are performed on several datasets to validate the effectiveness and efficiency of the proposed UEMCP under different metrics

**Strengths:**

Most existing multi-view clustering methods for large-scale data usually rely on anchor graphs to reduce the algorithmic complexity. The structure of the data can be represented by choosing $ m $ anchors to reflect the entire distribution for the dataset. The relation between data points and anchors can be built based on the anchor graph and the correlation tends to be stronger when a data point and anchor belong to the same category. Besides, a consensus data distribution shared by different views is assumed in our multi-view clustering setting. The anchor graph can be considered as the probability transition matrix between data points and anchors due to the non-negative properties and summarization being one for the row. The anchor selection and the probability transition matrix construction are separated from each other, which will inevitably affect the final performance. Different from the traditional strategy, the authors automatically learn anchors instead of simple sampling

**Weaknesses:**

. To study the influence of parameter $ \lambda $ on the final clustering performance, the authors perform experiments for investigation in the range of $ \{0.001, 0.005, 0.01, 0.05, 0.1, 0.5, 1\} $. According to Fig. 1, the authors find that choosing the proper parameter is crucial in improving the final clustering results. The parameter $ \lambda $ with too large or small values is not helpful in achieving the desired performance. However, the authors do not give detailed reason why choose $ \{0.001, 0.005, 0.01, 0.05, 0.1, 0.5, 1\} $ as the range in performing experiments for investigation.
2. The authors present the clustering results of the proposed UEMCP and compared methods on seven benchmark datasets. 'N/A' is adopted to show that the method encounters the out-of-memory issue on the dataset. According to Tables 2-5, we can draw conclusions. However, the authors do not bold the best clustering results in Tables 2-5 for the experiment part. Therefore, the authors are expected to bold the best clustering results in Tables 2-5 for the experiment part.
3. To validate the computational efficiency, we list the running time of our UEMCP on different datasets. As shown in Fig. 4, we observe that UEMCP needs relatively shorter running time on different datasets, which demonstrates the computational efficiency of our method. Though less running time is needed by BMVC, the simple procedure ignores to fully explore the information from multiple views, leading to relatively poor performance. Considering that the running time analysis is given in this paper as shown in the above, the authors are expected to list the memory of the adopted device in performing the experiment.
4. In the reference part, some publication name is called for short, i.e., Inf. Sci., and some publication is called for the whole name, i.e., Proceedings of the AAAI Conference on Artificial Intelligence. It is observed that these manners of publication names are not consistent. Then the authors should correct this issue and ensure that the forms of the references in this paper are consistent.

**Questions:**

1.The authors are expected to give detailed reason why choose $ \{0.001, 0.005, 0.01, 0.05, 0.1, 0.5, 1\} $ as the range in performing experiments for investigation.

2.The difference from existing jobs, especially those related to probability transitions, is not very prominent, It is recommended to supplement relevant work, highlight the differences of the paper

---

> ### Author Response · Authors · 2025-11-17
>
> Q1: The reason why 0.001,0.005,0.01,0.05,0.1,0.5,1 are selected as the range in performing experiments for investigation.
>
> A1: Thanks for the comment! As reviewer mentioned, the proper parameter is crucial in improving the final clustering results. The parameter \lambda with too large or small values is not helpful in achieving the desired performance. However, we do not give detailed reason why we choose 0.001,0.005,0.01,0.05,0.1,0.5,1 as the range in performing experiments for investigation. In the experiment, the reason why we choose {0.001, 0.005, 0.01, 0.05, 0.1, 0.5, 1} as a range is that these parameters represent different representative magnitudes in parameter selection. We will add such explanation regarding parameter selection for the camera-ready version.
>
> Q2: The best clustering results in Tables 2-5 are expected be bolded.
>
> A2: Good question! We should bold the best clustering results in Tables 2-5 for the experiment part as reviewer pointed. We will bold the best clustering results on Tables 2-5 in the experiment part for the camera-ready version.
>
> Q3: The memory of the adopted device in performing the experiment can be listed.
>
> A3: Thanks for the comment! As reviewer mentioned, we are expected to list the memory of the adopted device in performing the experiment considering that the running time analysis is given in this paper as shown in the above. The memory of the adopted device in performing the experiment is 8G and we will add such illustration in experiment part for the camera-ready version.
>
> Q4: The forms of the references should be consistent.
>
> A4: Good question! In the reference part, some publication names are called for short, i.e., Inf. Sci., and some publications are called for the whole name, i.e., Proceedings of the AAAI Conference on Artificial Intelligence. It is observed that these manners of publication names are not consistent as reviewer pointed. We will correct this issue and check the whole paper to ensure that forms of the references are consistent for the camera-ready version.
>
> Q5: The difference from existing jobs, especially those related to probability transitions, is not very prominent, It is recommended to supplement relevant work, highlight the differences of the paper.
>
> A5: Thanks for the comment! As reviewer recommended, the difference from existing jobs, especially those related to probability transitions, is not very prominent. Therefore, it is needed to the supplement relevant work and highlight the differences of the paper.
> Different from the related work based on probability transitions, i.e., FPMVS-CAG [a], the proposed method considers the interpretability from the probabilistic perspective for the anchor-based multi-view clustering, complicating the acquirement of $ k $ distinct connected components. It pays attentions to analyzing the relationship between the input data and the final clustering results in a unified manner. We have added the related analysis for the camera-ready version.
>
> [a] Fast Parameter-Free Multi-View Subspace Clustering With Consensus Anchor Guidance, IEEE Transactions on Image Processing, 2022.

---

### Official Review · Reviewer_zh1j · 2025-11-01

**Soundness:** 3
**Presentation:** 3
**Contribution:** 3
**Rating:** 8
**Confidence:** 5

**Summary:**

This paper introduces a novel anchor-based multi-view clustering method, UEMCP. The primary goal is to improve the interpretability of anchor-based clustering by framing it from a probabilistic perspective within an end-to-end learning model. The method aims to ensure structural consistency across views by learning a common transition probability matrix from data points to cluster categories in a single step.

**Strengths:**

The paper proposes an end-to-end framework to enhance the explanation ability of anchor-based multi-view clustering, which is a valuable goal. The method is designed to ensure the consistency of inherent structures across views. A key aspect is the use of a common transition probability matrix from anchors to categories, which guides the soft label assignment for data points within the learning framework. The experimental results on several datasets are strong and show a clear improvement over existing methods.

**Weaknesses:**

1.  The methodology is plagued by confusing and inconsistent notation. For example, in Eq.1, $P_v$ should be $d_v \times d$, but its constraint is $P_v^T P_v = I_m$ (it should be $I_d$). The variable $S$ is defined with different dimensions in Eq.1 ($m \times n$) and Eq.2 ($l \times n$). Similarly, $H$ has conflicting dimensions in Eq.2 ($c \times l$) and Eq.3 (implying $c \times m$). The dimension $l$ is introduced in Eq.2 without definition. These issues severely affect readability.
2. The paper presents a running time analysis, but the memory used for the experiments are not provided. Besides, the discussion of the running time is very brief. A more detailed comparison and analysis against the other methods are needed.
3. The formatting of the references is inconsistent and should be carefully checked throughout the paper.

**Questions:**

Could the authors provide a more detailed analysis of the running time results shown in Figure 4? Specifically, please elaborate on the comparative efficiency of UEMCP against the baselines.

---

> ### Author Response · Authors · 2025-11-17
>
> Q1: The methodology is plagued by confusing and inconsistent notation. These issues severely affect readability.
>
> A1: Thanks for the comment! As reviewer mentioned, the methodology is plagued by confusing and inconsistent notation, i.e., P_v should be d_v \times d and its constraint is P_v^TP_v=I_d. The variable S should be defined with dimensions m \times n and the dimensions in Eq. 1 and Eq. 2 should be consistent. H should have the dimension c \times m and then dimension l is not appeared. We have checked the whole paper to avoid the similar dimension inconsistency issues for the camera-ready version to improve the readability of the whole paper.
>
> Q2: The paper presents a running time analysis, but the memory used for the experiments are not provided. Besides, the discussion of the running time is very brief. A more detailed comparison and analysis against the other methods are needed.
>
> A2: Good question! Since the running time analysis is presented in this paper, we should give the memory of the used device in the experiment for the proposed method as reviewer pointed. The memory of the used device in the experiment for the proposed method is 8G and we will add this description for the camera-ready version. As reviewer mentioned, we should give more running time analysis based on the Figure 4 in the paper, i.e., the compared results between the proposed method and the other ones. Different the other methods, ours aims to increase the explanation ability of multi-view clustering built on anchor from the probabilistic perspective in an end-to-end manner. It can learn the common transition probability from data points to categories shared by different views with one step, which ensures the consistent inherent structures among these views. The adopted end-to-end manner of the proposed UEMCP as above mentioned can help reducing the running time and improving the efficiency in the experiment, which is different from some other compared methods. We will give the above detailed running time analysis for the camera-ready version.
>
> Q3: The formatting of the references is inconsistent and should be carefully checked throughout the paper.
>
> A3: Good question! The forms of the references in this paper should be guaranteed to be consistent as reviewer pointed and the we should carefully check the whole paper. We will check the whole paper to confirm that the forms of the references in this paper are guaranteed to be consistent for the camera-ready version.

---

### Official Review · Reviewer_baAD · 2025-11-06

**Soundness:** 3
**Presentation:** 3
**Contribution:** 3
**Rating:** 6
**Confidence:** 3

**Summary:**

This paper shows a new method termed Unified and Efficient Multi-view Clustering from Probabilistic perspective (UEMCP), which assigns the probabilistic meaning to the anchor graph and soft label of data points to increase the explanation ability of multi-view clustering model in an end-to-end manner. UEMCP is able to learn the common transition probability from data points to categories shared by multiple views with one step, which ensures the consistency of inherent structures for these views.

**Strengths:**

The authors propose a novel method termed Unified and Efficient Multi-view Clustering from Probabilistic perspective for increasing the explanation ability of multi-view clustering based on anchor from the probabilistic perspective. Besides, the expression is very clear with satisfied writing and novelty. With the guidance of the common transition probability matrix from anchor points to categories in the learning framework, the soft labels of data points are able to be achieved.

**Weaknesses:**

1. The authors can give the brief operations in Optimization part for Section 3.2. Then the main idea for optimization is more clear for readers.

2. To study the influence of parameter λ on the final clustering performance, the authors perform experiments for investigation in the range of {0.001, 0.005, 0.01, 0.05, 0.1, 0.5, 1}. The authors should explain why they choose such range in parameter selection.

3. The authors can give more details for running time analysis part in this work, i.e., the reason why the proposed UEMCP needs relatively less time in the experiment.

4. The authors should confirm the typo error and check the whole paper to avoid such issues for the paper.

**Questions:**

See the Weakness box.

---

> ### Author Response · Authors · 2025-11-17
>
> Q1: The authors can give the brief operations in Optimization part.
>
> A1: Thanks for the comment! As reviewer mentioned, we can give the brief operations in Optimization part for Section 3.2. Then the main idea of optimization is more clear for readers. We seek for the solution to the involved S-subproblem, A-subproblem, P_v-subproblem, H-subproblem, G-subproblem and \alpha_v-subproblem for the proposed UEMCP in the following. These subproblems are optimized with the other variables being fixed in the detailed procedure. We will add such brief operations in Optimization part for camera-ready version.
>
> Q2: Why choose {0.005, 0.01, 0.05, 0.1, 0.5, 1} as a range in parameter selection?
>
> A2: Good question! We perform experiments for investigation in the range of {0.001, 0.005, 0.01, 0.05, 0.1, 0.5, 1} to study the influence of parameter \lambda on the final clustering performance. Then we should explain why we choose such range in parameter selection. For the experiment, the reason why we choose {0.001, 0.005, 0.01, 0.05, 0.1, 0.5, 1} as a range is that these parameters represent different representative magnitudes in parameter selection. We will add such explanation regarding parameter selection for the camera-ready version.
>
> Q3: The authors can give more details for running time analysis.
>
> A3: Thanks for the comment! As reviewer mentioned, we can give more details for running time analysis part in this work, i.e., the reason why the proposed UEMCP needs relatively less time in the experiment. Our method aims to increase the explanation ability of multi-view clustering built on anchor from the probabilistic perspective in an end-to-end manner. It can learn the common transition probability from data points to categories shared by different views with one step, which ensures the consistent inherent structures among these views. The adopted end-to-end manner of the proposed UEMCP as above mentioned can help reducing the running time and improving the efficiency in the experiment. We will give the above detailed running time analysis for the camera-ready version.
>
> Q4: The authors are expected to confirm the typo error in the whole paper.
>
> A4: Good question! As reviewer mentioned, we are expected to confirm the typo error and check the whole paper to avoid such issues for the paper. We will check the whole paper to avoid the possible existing typo error for the camera-ready version.

---

### Meta-Review · Area_Chair_o2Vf · 2025-12-27

**Summary:**

UEMCP presents an anchor-based multi-view clustering framework with a clear probabilistic interpretation: it jointly learns a common transition probability from data points to categories in an end-to-end manner, improving interpretability, consistency across views, and efficiency on large datasets. All reviewers agree an acceptance to this paper.

**Reviewer Concerns:**

Almost all concerns were clarified in rebuttal and the reviewers raised no objections.

**Reviewer Scores:**

According to rebuttal, reviewers may maintain the current positive score.

---

### Decision · Program_Chairs · 2026-01-26

Accept (Poster)